psychology

impulsive action, replication, action control, gambling

**Author for correspondence:**
Charlotte Eben
e-mail: charlotte.eben@ugent.be

# A direct and conceptual replication of post-loss speeding when gambling

Charlotte Eben[1], Zhang Chen[1], Luc Vermeylen[1], Joël Billieux[2] and Frederick Verbruggen[1]

[1]Department of Experimental Psychology, Ghent University, Ghent, Belgium
[2]Institute of Psychology, University of Lausanne, Lausanne, Switzerland

 CE, 0000-0001-9423-1261

To investigate the response to suboptimal outcomes, Verbuggen *et al.* (Verbruggen F, Chambers CD, Lawrence NS, McLaren IPL. 2017 Winning and losing: effects on impulsive action. *J. Exp. Psychol.: Hum. Percept. Perform.* **43**, 147. (doi:10.1037/xhp0000284)) conducted a study in which participants chose between a gamble and a non-gamble option. The non-gamble option was a guaranteed amount of points, whereas the gamble option was associated with a higher amount but a lower probability of winning. The authors observed that participants initiated the next trial faster after a loss compared to wins or non-gambles. In the present study, we directly replicated these findings in the laboratory and online. We also designed another task controlling for the number of trials per outcome. In this task, participants guessed where a reward was hidden. They won points if they selected the correct location, but lost points if they selected the incorrect location. We included neutral trials as a baseline. Again, participants sped up after a loss relative to wins and neutral trials (but only with a response choice in neutral trials and a large sample size). These findings appear inconsistent with cognitive-control frameworks, which assume that suboptimal outcomes typically lead to slower responses; instead, they suggest that suboptimal outcomes can invigorate behaviour, consistent with accounts of frustrative non-reward and impulsive action.

## 1. Introduction

Cognitive control theories assume that the cognitive system monitors ongoing actions and their outcomes, and adjusts task settings and response strategies when the action outcomes are undesired or suboptimal (e.g. an error leading to punishment or the absence of a reward). It has been argued that failures to adjust behaviour and learn from (negative) past experiences is central to several mental disorders and behavioural problems, including substance abuse and

addictive disorders [1–4]. It is, therefore, important to understand how behaviour is adjusted following suboptimal outcomes.

In the present study, we further examined how negative outcomes influence performance on subsequent trials. Between-trial adjustments following suboptimal outcomes are typically associated with changes in response speed, and slowing in particular. For example, participants often slow down after committing an error [5,6]. However, it is still debated why behavioural responses are slowed. Cognitive-control theories attribute such slowing to changes in response settings, leading to more cautious behaviour (i.e. a shift from speed towards accuracy [7]). In recent years, it has become clear though that post-error slowing is not entirely due to such strategic task adjustments. After all, response slowing is not always associated with an increase in accuracy, as one would expect if the slowing is due to a more cautious response style. Some studies even found increased error rates following errors [6,8,9]. Therefore, Notebaert *et al.* [10] proposed an orienting account for post-error slowing. Errors are typically relatively rare in an experiment. Previous work in the attention literature suggests that rare events can slow performance as they orient attention away from the task. Therefore, Notebaert *et al.* argued that infrequent errors may also orient attention away from the focal task, causing longer response latencies (and increased error rates) on subsequent trials. Consistent with this idea, they found that errors no longer produced post-error slowing when errors were more frequent than correct responses [10]. Recently, Wessel [11] integrated both views in his 'adaptive orienting' account. This account assumes that all unexpected outcomes lead to re-orienting of attention to the source of the unexpected outcome. In the case of errors or other suboptimal events, this orienting phase is subsequently followed by a task-adjustment phase [11]. When there is enough time available after a suboptimal outcome, these adjustments can be adaptive, leading to increased accuracy. However, if there is little time available, there is not enough time for adjustments, leading to decreased accuracy.

Despite the different opinions on what is causing sequential (between-trial) effects, most researchers now seem to agree that suboptimal outcomes such as errors lead to slower responses (and adjustments when sufficient time is available). However, recent work on sequential effects after losses in gambling tasks questions whether slowing after suboptimal outcomes is a general phenomenon after all. Previous work suggests that similar neural networks (typically associated with performance monitoring) are activated by errors in choice tasks and losses in gambling tasks [12]. Studies measuring facial electromyographic activity (often used as a marker of emotional responses) after errors and losses also indicate overlap [13–15]. Thus, based on such findings, it has been argued that losses and errors (which are both suboptimal or undesirable outcomes) are processed in a similar way (e.g. [12]). However, one should be aware of potential reverse inference problems (e.g. [16]). Furthermore, despite this (purported) overlap, speeding (instead of slowing) has been observed after suboptimal outcomes (losses) in gambling tasks. For example, Verbruggen *et al.* [17] used a simple gambling task in which participants choose between a non-gamble option with a guaranteed (low) amount of points and a gambling option with a higher amount of points but a lower probability of winning. In all five experiments of the study, participants initiated the next trial (start response time (start RT)) faster after a gambled loss compared to the non-gamble baseline or a gambled win. Similar speeding after losses has been observed in other gambling tasks as well [18,19]. In Verbruggen *et al.* [17], this post-loss speeding was most pronounced when the potential win amount was high (and hence when the probability of a loss was high, as high win amount was associated with low win probability to keep the expected value matched between the gamble and the non-gamble option). Furthermore, losses also influenced response latencies in a perceptual decision-making task that was intermixed with the gambling task. Combined, these findings appear inconsistent with control and orienting accounts. Instead, the findings of these experiments indicate that losses (or failures to obtain reward) can invigorate subsequent behaviour, or as the authors of the original study put it, 'impulsivity' rather than 'restraint'.

Verbruggen *et al.* [17] speculated that the post-loss speeding might be due to 'frustration' (i.e. a negative affective state induced by the failure to obtain a reward or by the blockage of a desired goal) or 'regret' (the realization that another choice would have produced a more desired outcome). After all, previous work on humans and non-human animals suggest that 'frustration' or 'regret' might invigorate subsequent behaviour [20–22]. For example, Amsel [20] observed that hungry rats who were trained to obtain food in two runways ran faster in the second runway when they failed to obtain (the expected) food in the first runway. More generally, several theoretical frameworks assume an influence of such negative affective states on actions (for a review see [23]). For example, Frijda [24] suggests that events are appraised by individuals as pleasant or unpleasant, triggering states of action readiness (i.e. a state to change or sustain the individual's relation to the event). According to this view, negatively appraised outcomes will promote impulsive actions.

According to Frijda [24], impulsive actions are affective in nature. Yet, several researchers have argued that emotions might also influence cognitive control and strategic performance adjustments [25–29]. For example, Riesel *et al.* [28] found that post-error slowing was more pronounced when errors were punished, and they attributed this effect to negative affective states induced by the punishment. Furthermore, van Steenbergen *et al.* [29] found that randomly presented monetary gains as a feedback in a flanker task led to reduced subsequent response caution.

Thus, it appears that affective states play a role in both the origin and control of impulsive actions. Saunders & Inzlicht [30] proposed the 'shifting priorities' model to explain how suboptimal outcomes such as an error or a loss may result in such distinct after-effects (i.e. impulsive action versus restraint). According to these authors, experiencing a long period of unrewarded control processes might lead to an attempt to return to 'cognitive comfort' by the individual (i.e. a state which is characterized by a low level of negative affect). Starting from a discrepancy between the current state and a desired state, contextual and environmental factors determine the best strategy to achieve this. After an error in a controllable task, increasing cognitive control might lead to cognitive comfort by reducing subsequent errors. By contrast, when individuals have no control and perceive the situation as unsolvable, they might experience fatigue and, therefore, invest reduced or even no cognitive effort [22,31]. Uncontrollable situations may even lead to impulsive actions when behaviour is invigorated without further adjustments of task settings. Thus, the authors suggested that increased vigour versus restraint after negative outcomes depends on the task context [30].

The present study aims to further clarify when a loss leads to invigoration of behaviour and how task-specific the effects observed by Verbruggen *et al.* [17] are. In order to achieve this objective, we conducted four experiments using different tasks in which participants can win or lose points. Consistent with the original study, these points were converted into real money at the end of the experiment. In Experiment 1A and 1B we directly replicated the study by Verbruggen *et al.* [17] by using the same gambling task. In Experiment 2, we used a modified version of the 'doors task' originally used by Dunning & Hajcak [32]. In the original study [32], participants had to guess behind which of the two presented doors a reward was hidden. In the present study, the task was modified to create a neutral (non-gamble) baseline. After all, the literature suggests that both wins and losses can influence behaviour. To disentangle post-loss speeding (shorter latencies after a loss) and post-reinforcement pause (longer latencies after reward [33]), we introduced a third 'non-gamble' door. On some trials, participants had to select this door (indicated by a colour cue), and they could not win or lose any points. However, we were not able to replicate the post-loss speeding or post-win slowing with this set-up. Therefore, in Experiment 3, we further modified this task and used a set-up which looked like playing cards. In this set-up we introduced a choice element for the non-gamble trials to enable participants deciding between two keys instead of one key to continue. Post-loss speeding was successfully observed with this modified version of the task. As discussed below, this experiment allowed us to rule out an alternative 'attentional orienting' explanation for our findings.

# 2. Experiment 1A

The aim of the first experiment was to replicate the findings of Verbruggen *et al.* [17]. We thus conducted a laboratory replication in Experiment 1A with the exact same task.

## 2.1. Method

### 2.1.1. Participants

Twenty-two students (*range* 18–27 years; $M = 21$ years; 17 female) from Ghent University were tested individually. Two participants were excluded due to a low amount of trials per condition (see below). Our power calculation was based on the smallest effect size of the original study (Cohen's *d*: 0.8) for a paired two-tailed *t*-test, alpha = 0.05 and intended power of 0.95. Written informed consent was obtained. The study was approved by the local research ethics committee at the Faculty of Psychology and Educational Science of Ghent University. This applies to all experiments included in the current study.

### 2.1.2. Apparatus and stimuli

The experiment ran on Windows desktop PC with PsychoPy 3.0 [34]. Responses were recorded with a keyboard. Participants had to press the left or right arrow key to choose either option. In contrast to the

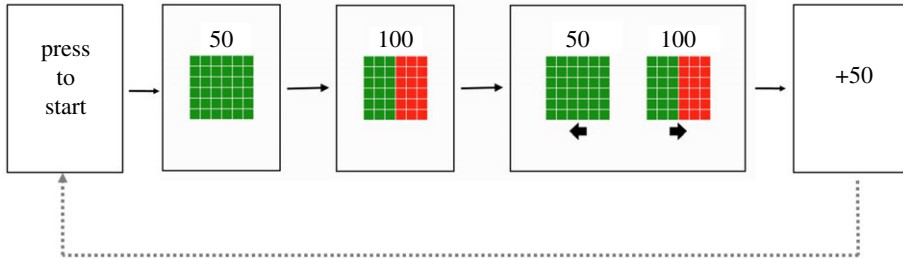

**Figure 1.** The trial procedure of Experiment 1A.

original study, in each trial we presented two sets of smaller red and green squares (instead of red and green pie charts). This change had pure practical purposes as it was easier to program squares instead of pie charts in PsychoPy. The relative proportion of green squares indicated the probability to win (figure 1). The amount participants could win was presented centrally above the squares (see figure 1; font: Arial; font size: 25 point; font colour: white). The first option always represented a non-gamble option. If participants selected this option in the selection phase, they always received the (guaranteed) amount that was presented above the squares. The non-gamble option consisted entirely of smaller green squares. The second option represented the gamble option. When participants selected this option, they could win an amount that was higher than the amount associated with the non-gamble option, but the probability of winning it was lower than 1. The exact probability of winning was indicated by the proportion of green smaller squares. Thus, the more green squares, the higher the probability to win. The red squares (proportion of red squares = $1 - p(\text{win})$) were always on the right side of the bigger square. The options were presented against a grey background. In the selection phase (figure 1) arrows under the squares indicted which key should be pressed for which option.

We randomized the amount and the probability to win across trials. The amount associated with the non-gamble option was 20, 30, 40 or 50. The amount associated with the gamble was 1.5, 2, 3 or 4 times higher than the non-gamble amount. This resulted in 16 possible combinations. The probability of winning varied between 0.67, 0.50, 0.33 and 0.25 and was adjusted to keep the expected value of the gamble and the non-gamble the same (for further details see [17] and the appendix of that study).

### 2.1.3. Procedure

The trial course is depicted in figure 1. Each trial started with the message 'Please press a key to start the next game.' After participants had pressed a key, the non-gamble option with the guaranteed amount was presented in the centre of the screen. After 1s, we presented the gamble option with the possible win amount for one second. After this, both options were presented together (the selection phase of the trial), one on the left and one on the right. At this point, participants had to choose the non-gamble option or the gamble option by pressing the corresponding left or the right arrow key. The location of the non-gamble and gamble option was also randomized. After participants executed a choice response, we showed the outcome of the participant's choice. If they had selected the gamble, the computer indicated whether they had won the points indicated with the squares (gambled win; e.g. 'outcome = 200 points') or not (gambled loss; 'outcome = 0 points'). To determine the outcome of a gamble, the computer selected a random number between 0 and 1 on each trial, and participants had won the gamble if the selected number was smaller than $p_{\text{win}}$. If they had selected the non-gamble option, participants always received the amount associated with the non-gamble option (e.g. outcome = 50 points). After 1 s, the next trial started by displaying the message 'Press a key to start the next trial.'

The gambling task consisted of 256 trials. For most participants, the experiment lasted 20–30 min. Participants were told that they are allowed to take a short 'mini-break' between trials, as there were no fixed breaks after a set number of trials. Consistent with the original study [17] we made choices consequential: at the end of the experiment, the computer randomly selected the outcomes of 10 trials. The sum of these trials was converted into real money: for every 100 points, participants gained 1 Euro extra. The maximum additional payout was 5 Euro (range: 0–5 Euro). Participants were informed about the pay-off structure at the beginning of the experiment. At the end of the experiment, all participants filled in the UPPS-P short questionnaire [35]. This impulsivity questionnaire was included as part of a larger individual-differences project across studies, and not further analysed for this specific study.

**Table 1.** Inferential statistics Experiment 1A. diff, difference; CI, confidence interval (95%); BF, Bayes Factor 10; $g_{av}$, Hedge's average $g$.

| | diff | lower CI | upper CI | d.f. | $t$ | $p$-value | BF | $g_{av}$ |
|---|---|---|---|---|---|---|---|---|
| **start RT** | | | | | | | | |
| non-gamble versus loss | 158.41 | 93.64 | 223.18 | 18 | 5.139 | <0.001 | 390.53 | 0.660 |
| non-gamble versus win | 87.58 | 35.18 | 139.99 | 18 | 3.511 | 0.002 | 16.82 | 0.349 |
| loss versus win | −70.83 | −133.39 | −8.26 | 18 | −2.378 | 0.029 | 2.20 | 0.325 |
| **choice RT** | | | | | | | | |
| non-gamble versus loss | 33.93 | −4.98 | 72.84 | 18 | 1.832 | 0.084 | 0.95 | 0.235 |
| non-gamble versus win | −8.18 | −47 | 30.64 | 18 | −0.443 | 0.663 | 0.26 | 0.047 |
| loss versus win | −42.11 | −92.36 | 8.14 | 18 | −1.761 | 0.095 | 0.86 | 0.288 |
| **probability to gamble** | | | | | | | | |
| non-gamble versus loss | −0.02 | −0.08 | 0.04 | 18 | −0.582 | 0.568 | 0.28 | 0.078 |
| non-gamble versus win | 0.04 | −0.03 | 0.12 | 18 | 1.243 | 0.230 | 0.46 | 0.192 |
| loss versus win | 0.06 | 0.00 | 0.12 | 18 | −2.110 | 0.049 | 1.44 | 0.260 |

### 2.1.4. Analyses

All data processing and analyses were completed with R [36]. All raw data files and R scripts of all experiments can be found on OSF [37]. We excluded participants who had too few trials (less than five trials in one or more conditions). In the analyses of sequential trials, we distinguished between trials that followed a non-gamble trial (our baseline), trials that followed a gambled win, and trials that followed a gambled loss. For each trial type, we calculated how fast participants started the next trial (start RT), the probability to gamble, and the latency of choosing between the gamble and the non-gamble option (choice RT). We excluded trials on which start RT was above 5000 ms, or choice RT (i.e. the left/right arrow response) was above 2500 ms. The analyses focused on the effect of the outcome of the previous trial; therefore, we also excluded the first trial of the experiment. This resulted in a data exclusion of 2.0%. The trial exclusion criteria were determined before data collection and based on the exclusion criteria used by Verbruggen *et al.* [17].

Inferential statistics are presented in table 1. As we were trying to replicate the previous results, and given our main research question, we were mostly interested in the differences with the non-gamble baseline (i.e. the omnibus ANOVA cannot inform us whether losses, wins, or both are influencing response speed). Therefore, we directly conducted the (planned) pairwise comparisons. The three trial types (i.e. trials following a non-gamble, gambled win and gambled loss) were compared with two-tailed *t*-tests and their Bayesian equivalent. The Bayes Factor BF10 (calculated with the BayesFactor package v. 0.9.12–4.2 in R) quantifies the evidence for the alternative hypothesis against the null hypothesis. We used the default prior widths (i.e. the Cauchy prior with a width of 0.707) as defined by the BayesFactor package. For the pairwise comparisons, Hedge's $g_{av}$ is the reported effect size measure [38]. We used the Holm–Bonferroni correction for multiple comparisons.

## 2.2. Results and discussion

Consistent with Verbruggen *et al.* [17], we found that the start RT was influenced by the outcome of the previous trial: participants started the next trial faster after a loss ($M = 499$ ms; s.d.= 202 ms) than after a non-gamble trial ($M = 658$ ms; s.d.= 268 ms) or a win ($M = 570$ ms; s.d.= 224 ms). All differences were statistically significant (table 1, but see figure 5). Thus, we replicated the main findings of Verbruggen *et al.* [17]. Start RT was generally shorter after a gamble than a non-gamble. This could be caused by increased arousal associated with gambling. Importantly, the speeding was most pronounced after a loss. This is consistent with the idea that negative outcomes can invigorate subsequent behaviour.

Probability of gambling was numerically lower after a win ($p_{gamble} = 0.42$; s.d.= 0.23) than after a loss ($p_{gamble} = 0.48$; s.d= 0.22) or a non-gamble ($p_{gamble} = 0.47$; s.d.= 0.21). However, these differences were not statistically significant after correction for multiple comparisons. There were also numerical choice

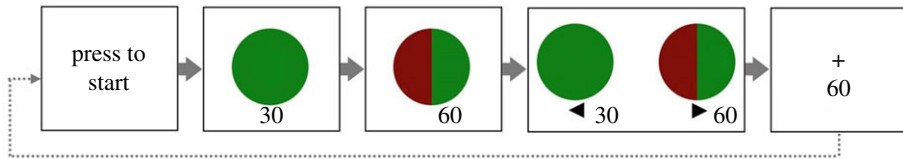

**Figure 2.** The trial procedure of Experiment 1B, which is similar to the original task by Verbruggen *et al.* [17].

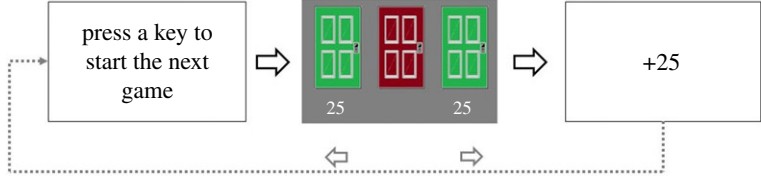

**Figure 3.** The trial procedure of Experiment 2. In this gamble trial, the participant selected the correct door.

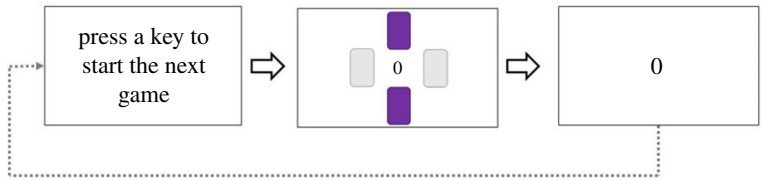

**Figure 4.** Trial procedure Experiment 3. This represents a non-gamble trial.

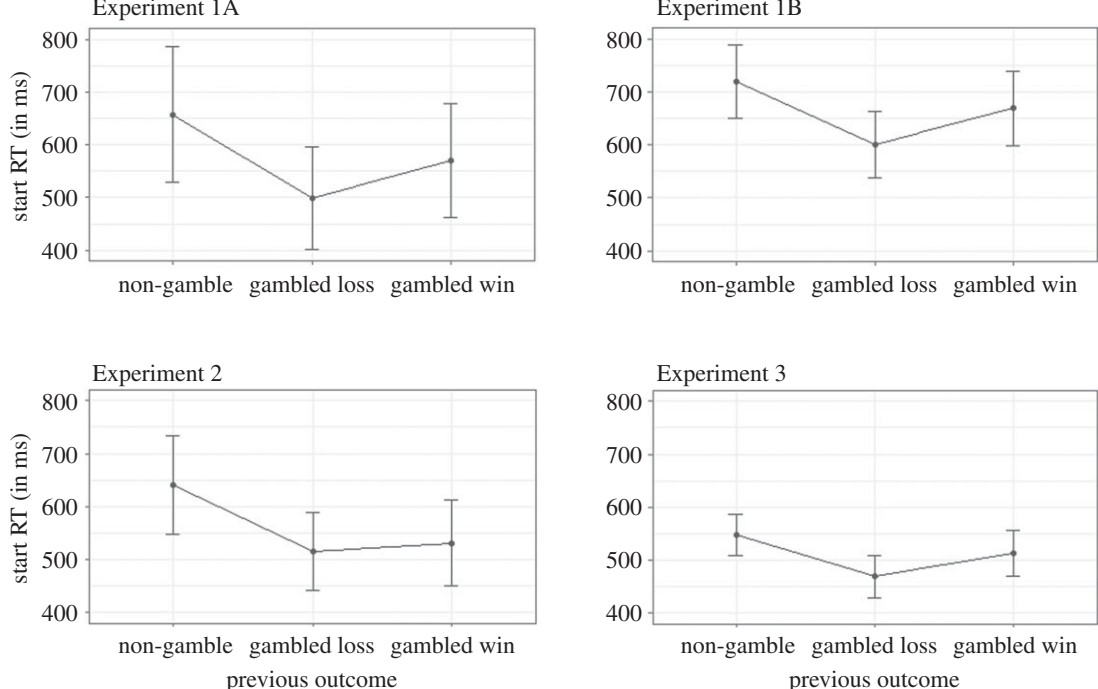

**Figure 5.** Start RT as a function of previous outcome for all experiments. The error bars reflect within-subject confidence interval.

RT differences between trials following a loss ($M = 579$ ms; s.d.= 114 ms), a non-gamble ($M = 613$ ms; s.d.= 168 ms), or a win ($M = 621$ ms; s.d. = 172 ms). But again, these differences were not significant.

## 3. Experiment 1B

After replicating the main start RT results of Verbruggen *et al.* [17] in the laboratory, we aimed to replicate it again using an online sample. Online testing has become increasingly popular in recent years as it

allows testing (relatively quickly) big samples with a more heterogeneous population than undergraduate students of a particular university [39]. Usually, crowdsourcing platforms like MTurk or Prolific are used for online testing. Here, we decided to acquire participants via Prolific, as this platform was specifically developed for scientific research (for a discussion see [39]). Furthermore, Prolific seems to produce high-quality data and well-known effects were already replicated via online studies via this platform [40].

## 3.1. Method

### 3.1.1. Participants

One hundred participants (recruited via Prolific) completed the whole online experiment (49 females; age $M = 29.5$ years, s.d. = 10 years; *range* = 18–67, but data of two participants are missing). We had to exclude 14 participants due to too few trials per condition (see Experiment 1A), resulting in 86 datasets for analysis. Participants agreed to the consent form before starting the experiment. In theory, participants from all over the world can sign up for our online studies via Prolific. However, as our instructions were in English (in all experiments, including the laboratory-based studies), we added knowledge of English as a criterion for Experiment 1B. For the first language in this study Prolific provided us with the following data: 38% of participants were native English, 16% Polish, 7% Portuguese, 6% Italian, 5% German, 4% Estonian, 4% Greek, 4% Spanish, 2% Dutch, 2%, Russian, 1% Czech, 1% French, 1% Indonesian, 1% Japanese, 1% Serbian, 1% Slovenian, 1% Swedish and 1% Tagalog-Filipino speaking. For the nationality in this study, 22% were English (UK) participants, 17% Polish, 8% Portuguese, 6% Italian, 6% Australian, 4% Greek, 4% Estonian, 3% German, 3% Mexican, 2% Austrian, 2% Canadian, 2% Latvian, 2% Finnish, 2% American, 2% New Zealand and 1% each Belgian, Czech, Dutch, French, Indonesian, Iranian, Israeli, Japanese, Luxembourgish, Philippines, Serbian, Slovenian, Spanish, Swedish and Vietnamese. In addition to the 100 participants who completed the experiment and got paid, 33 participants signed up for the experiment on Prolific but did not start or complete it, and two participants were excluded as their completion time was under 15 min whereas the average time taken was 29 min. All data were acquired on 5 June 2019. Participants had to enter their nationality manually, as this ensured that no bots could complete the task.

### 3.1.2. Apparatus and stimuli

The experiment only ran on desktop computers and laptops, with Chrome or Mozilla Firefox installed (the experiment runs without any problems in these two browsers [41]). The experiment was programmed in jsPsych, which has a comparable accuracy in response time measurements as standard laboratory software [41]. Keyboards were used to register responses. The only difference to Experiment 1A was the presentation of the non-gamble and gamble option. Instead of squares on each trial, we presented two pie charts (as in the original study of Verbruggen *et al.*). The green and red parts of the pie charts represented the probability to win. The amount was shown centred under the pie charts (font: Arial; font size: 25 point; font colour: black). The red part of the pie charts was always on the right side. The options were presented against a white instead of a grey background. Randomization of the amounts and the probabilities was the same as in Experiment 1A.

### 3.1.3. Procedure

The trial procedure is depicted in figure 2. The procedure was strictly similar to Experiment 1A except for the payout: for every 100 points, participants got £1 extra. The maximum additional payout was £3 (range: £0–3). Participants were informed about the pay-off structure at the beginning of the experiment.

### 3.1.4. Analyses

As in Experiment 1A, all data processing and analyses were completed with R [36]. Due to a failure in data online recording for eight trials across participants, we excluded trials in which the previous outcome was not known. Combined with the other exclusion criteria (see Experiment 1A), this resulted in a data exclusion of 4.5%.

**Table 2.** Inferential statistics Experiment 1B. diff, difference; CI, confidence interval (95%); BF, Bayes Factor 10; $g_{av}$, Hedge's average $g$.

| | diff | lower CI | upper CI | d.f. | $t$ | $p$-value | BF | $g_{av}$ |
|---|---|---|---|---|---|---|---|---|
| **start RT** | | | | | | | | |
| non-gamble versus loss | 118.56 | 79.98 | 157.14 | 85 | 6.110 | <0.001 | $4.13 \times 10^5$ | 0.386 |
| non-gamble versus win | 50.44 | 11.73 | 89.15 | 85 | 2.591 | 0.011 | 2.73 | 0.155 |
| loss versus win | −68.12 | −106.97 | −29.27 | 85 | −3.487 | 0.001 | 29.60 | 0.220 |
| **choice RT** | | | | | | | | |
| non-gamble versus loss | 3.69 | −17.14 | 24.52 | 85 | 0.352 | 0.725 | 0.13 | 0.018 |
| non-gamble versus win | −28.87 | −52.72 | −5.02 | 85 | −2.406 | 0.018 | 1.80 | 0.132 |
| loss versus win | −32.56 | −57.77 | −7.35 | 85 | −2.568 | 0.012 | 2.60 | 0.153 |
| **probability to gamble** | | | | | | | | |
| non-gamble versus loss | −0.08 | −0.11 | −0.05 | 85 | −4.885 | <0.001 | 3282.98 | 0.372 |
| non-gamble versus win | −0.05 | −0.08 | −0.01 | 85 | −2.574 | 0.012 | 2.63 | 0.208 |
| loss versus win | 0.03 | 0.01 | 0.06 | 85 | 2.466 | 0.016 | 2.06 | 0.142 |

## 3.2. Results and discussion

Consistent with Experiment 1A and the original study of Verbruggen *et al.* [17], start RT was influenced by the outcome of the previous trial: participants started the next trial faster after a loss ($M = 601$ ms; s.d. $= 291$ ms) than after a non-gamble trial ($M = 719$ ms; s.d. $= 321$ ms) or a win ($M = 668$ ms; s.d. $= 326$ ms). All differences were statistically significant (table 2, but see figure 5).

The probability to gamble after a loss ($p_{gamble} = 0.45$; s.d.$=0.22$) was higher than after a non-gamble ($p_{gamble} = 0.37$; s.d. $=0.21$) or after a win ($p_{gamble} = 0.42$; s.d.$=0.25$). Moreover, participants gambled more after a win than after a non-gamble. All differences were significant.

For the choice RT, we found no significant difference between trials following losses ($M = 674$ ms; s.d. $= 196$ ms) and non-gambles ($M = 677$ ms; s.d. $= 207$ ms). The differences between wins ($M = 706$ ms; s.d. $= 229$ ms) and the other two trial types were significant, indicating longer choice RT for trials following a win than trials following a non-gamble or a loss (table 2).

# 4. Experiment 2

The study by Verbruggen *et al.* [17] and Experiment 1A and 1B of the present study suggest that failures to obtain a reward in a gambling task can invigorate subsequent actions and increase probability of gambling (at least in Experiment 1B). In other words, negative outcomes do not always result in slowing or more cautious behaviour. Yet, we cannot completely rule out an attentional orienting account. This account assumes that unexpected outcomes lead to re-orienting attention to the source of this unexpected outcome. If participants gamble in most of the trials, in these gamble trials winning is less likely than losing (in our recent experiments the probability of winning was 46%); therefore slower responses after gambled wins could be due to such a re-orienting response. It is difficult to rule out this interpretation using the original task of Verbruggen *et al.* First, overall probabilities of winning cannot be controlled as the task is a free-choice task. Furthermore, an analysis of the effect as a function of win probabilities is also not straightforward: as the expected value is kept the same, the probability to win is necessarily confounded with the win amount.

To further investigate the underlying causes of post-loss speeding and/or post-win slowing, we developed another task to rule out the idea that our findings depended on the relative frequency of the events (as suggested by attentional orienting accounts [10,11]). Furthermore, by using a different task, we could also determine if post-loss speeding effect is task-specific or not.

To test the orienting account, we used a variant of the so-called doors task of Dunning & Hajcak [32]. In the original study, participants had to guess behind which of the two presented doors a reward is hidden. A cue indicated whether 0, 1 or 2 doors would contain the reward. Dunning & Hajcak [32] measured whether an error-related negativity (ERN) could be elicited by a predictive cue indicating a loss. The ERN is a negative deflection in the event-related potential (ERP) which can be observed in electroencephalography (EEG) measurements after the event of an error. These authors found that the cue indicating a loss elicited an ERN, which suggests that the component is sensitive to the first indication of a suboptimal outcome. In our version of the task, we also introduced non-gambling trials on which participants could not win or lose points. An advantage of this doors task is that the number of trials for each outcome (gambled win, gambled loss, non-gamble) can be predetermined.

## 4.1. Method

### 4.1.1. Participants

Twenty-four students (17 females; age $M = 21$ years, s.d. $= 2.8$, *range* $= 18$–29; two participants did not indicate their age and gender) from Ghent University participated for monetary compensation (5 Euro plus the reward they won). The target sample was determined before data collection based on Experiment 1A ($N = 19$) in which large effects for post-loss speeding were found (Cohen's $d = 0.8$).

### 4.1.2. Apparatus and stimuli

The experiment ran on a Windows computer using PsychoPy 3.0 [34]. We used a QWERTY keyboard to register responses. On each trial, participants saw three doors. On one-third of the trials, the middle door was green and the outer doors were red. The number '0' was presented below the green door. On these 'non-gamble' trials, participants simply had to press the space bar and they could not win or lose any points. On the remaining two-thirds of the trials, the outer doors were green and the middle door was red. These were the 'gamble' trials.

### 4.1.3. Procedure

The trial procedure is depicted in figure 3. Every trial started with a message 'Press a key to start the next gamble.' We displayed this message until participants pressed one of the response keys. After this, three doors were presented. On two-thirds of the trials (indicated by two green doors, presented left and right from the red middle door), participants had to press the left or the right arrow key to guess behind which door the reward was hidden. They were told that if they picked the 'correct' door, they would win the amount shown on the doors; but if they picked the 'incorrect' door, they would lose that amount. The outcome of a trial was actually predetermined. If the trial was predetermined to be a 'win', the participants were told that they had guessed correctly, whereas we told them they guessed incorrectly if the trial was predetermined to be a loss. The numbers '5', '25' or '50' were displayed below the outer doors to indicate the amount in cents that participants could possibly win (if they 'guessed correctly') or lose (if they 'guessed incorrectly'). On the remaining one-third of the trials, participants had to pick the door in the middle (by pressing the space bar). On these non-gamble trials, participants could not win or lose points. These trials were our baseline. The amounts and gambles were pseudo-randomized, and there was an equal number of wins (1/3), losses (1/3) and non-gambles (1/3). Note that in combination with amount, this resulted in seven possible outcomes: −50 (1/9), −25 (1/9), −5 (1/9), 0 (3/9), +5 (1/9), +25 (1/9), +50 (1/9).

The amount the participants could possibly win varied between 5 cents, 25 cents and 50 cents. The participants started with an amount of 2 Euro and were told that depending on their performance they could either win or lose. As outcomes were fixed, the total amount at the end of the experiment was always 2 Euro.

**Table 3.** Inferential statistics Experiment 2. diff, difference; CI, confidence interval (95%); BF, Bayes Factor 10; $g_{av}$, Hedge's average $g$.

| | diff | lower CI | upper CI | d.f. | $t$ | $p$-value | BF | $g_{av}$ |
|---|---|---|---|---|---|---|---|---|
| **start RT** | | | | | | | | |
| non-gamble versus loss | 124.95 | 64.70 | 185.19 | 23 | 4.290 | <.001 | 110.33 | 0.624 |
| non-gamble versus win | 109.92 | 55.95 | 163.89 | 23 | 4.213 | <0.001 | 93.06 | 0.533 |
| loss versus win | −15.03 | −58.48 | 28.42 | 23 | −0.715 | 0.482 | 0.27 | 0.082 |
| low versus medium | −12.67 | −48.19 | 22.84 | 23 | −0.738 | 0.468 | 0.28 | 0.075 |
| low versus high | −25.16 | −56.17 | 5.85 | 23 | −1.679 | 0.107 | 0.73 | 0.125 |
| medium versus high | −12.49 | −59.08 | 34.108 | 23 | −0.55 | 0.585 | 0.25 | 0.068 |
| **choice RT** | | | | | | | | |
| non-gamble versus loss | 21.23 | 1.28 | 41.19 | 23 | 2.201 | 0.038 | 1.62 | 0.110 |
| non-gamble versus win | −10.80 | −36.57 | 14.97 | 23 | −0.0867 | 0.395 | 0.30 | 0.054 |
| loss versus win | −32.03 | −54.78 | −9.28 | 23 | −2.912 | 0.008 | 5.94 | 0.164 |

### 4.1.4. Analyses

As in Experiment 1A and 1B, all data processing and analyses were completed with R [36]. We conducted the same analyses as in the previous experiments. Combined with the other exclusion criteria (see Experiment 1A), this resulted in a data exclusion of 2.3%. Here, we also examined how the amount won or lost on the previous trial modulated start RT differences. More specifically, we ran a $2 \times 3$ ANOVA with the independent variables previous outcome (win versus loss) and previous amount (high versus medium versus low) on the start RT. As the amount of the non-gamble trials always was 0 we excluded trials in which the previous trial was a non-gamble trial from these analyses. (Note that we did not perform these amount analyses for choice RT as it would require a 2 (outcome) by 3 (previous amount) by 3 (current amount) ANOVA, and we did not have enough trials for this.)

## 4.2. Results and discussion

The start RT was influenced by the outcome of the previous trial. Participants started the next trial faster after a loss ($M = 516$ ms; s.d. $= 174$ ms) and after a win ($M = 531$ ms; s.d. $= 192$ ms) than after a non-gamble trial ($M = 641$ ms; s.d. $= 222$ ms). The differences with the non-gamble baseline were significant, but unlike Experiments 1A and 1B, there was no statistically significant start RT difference between trials following a loss and a win (table 3, but see figure 5).

The $2 \times 3$ ANOVA with previous outcome and previous amount as independent variables revealed no significant interaction between previous outcome and previous amount, $F(2, 46) = 1.76$, $p = 0.183$, nor a main effect of previous amount or previous outcome, $Fs < 1$, $p > 0.378$.

Choice RT was shorter after losses ($M = 700$ ms; s.d. $= 188$ ms) than after wins ($M = 732$ ms; s.d. $= 197$ ms), $p < 0.05$. The difference between non-gamble trials ($M = 721$ ms; s.d. $= 193$ ms) and wins was not significant. The difference between losses and non-gamble trials was also not significant after correction for multiple comparisons (table 3).

In sum, we observed shorter start RTs after 'gamble' trials (i.e. trials on which participants had to guess under which door a reward was hidden) compared with 'non-gamble' trials on which no points could be won or lost. As noted above, this general speeding might be due to increased arousal on such gamble trials. However, we did not find any differences between losses and wins. Therefore, we ran a third experiment in which we made some modifications to the procedure, which are explained in the next section.

## 5. Experiment 3

There are (at least) three possible explanations for the absence of a difference between losses and wins in Experiment 2. We tried to address these issues in Experiment 3.

First, it is possible that the effect of post-loss speeding is smaller in the doors task, and hence, the power of the experiment was not sufficient to detect post-loss speeding. Therefore, we ran Experiment 3 online with a bigger sample size. Second, in the original gambling task, selecting the non-gamble option involved a choice, whereas no such choice element was required in Experiment 2 (participants simply had to press the space bar when the middle door was green). In the new version of the task (used in Experiment 3), we used a set-up looking like playing cards with two horizontal cards and two vertical cards. The horizontal cards were used on gambling trials (similar to Experiment 2), whereas the vertical cards were used on non-gambling trials. On these non-gambling trials, participants had to select one of the two vertical cards to continue the trial by pressing the up or down arrow key. Thus, the non-gambling trials also involved some choice element (just like the gambling trials), although the choice was non-consequential (i.e. regardless which card they picked, they could not win or lose any points). Third, overall task engagement might have been low in Experiment 2 as participants could have figured out that they had no influence on the task at all. Therefore, we varied the colour of the gamble trial cards in Experiment 3. The cards could be either blue, yellow, orange or green whereas the non-gamble cards were always associated with purple. Participants often have a tendency to look for hidden rules or patterns in tasks or games [42,43]. Note that participants also might have looked for rules in Experiment 2 and as soon as they realized that they have no influence on the outcome, task engagement dropped and the outcome no longer mattered. To test this idea, we added an exploratory analysis testing the first and the second part of Experiment 2. We found a numerical post-loss speeding effect in the first part but not in the second part. Therefore, we speculated that task engagement might increase if we introduced an extra feature (in this case, the colour of the cards), even when this feature was irrelevant.

## 5.1. Method

### 5.1.1. Participants

One hundred participants were recruited via Prolific and tested online (59 female; $M = 35$ years; s.d. = 13 years; *range* 18–67 years). As in Experiment 1B, participants had to enter their nationality manually to ensure that no bots could participate in the experiment. In the experiment we decided to only test a native English speaking sample. In this sample were 63% English (UK) participants, 16% American, 10% Australian, 4% Canadian, 3% New Zealand, and 1% each Iraqi, Irish, Philippines and Spanish. We had to exclude two participants as the majority of trials were not recorded for these participants due to technical issues. Moreover, we excluded one participant as their data contained some aberrant negative choice latencies (indicating timing issues). Therefore, we analysed 97 datasets. Participants had to agree to the consent form before starting the experiment. In addition to the 100 participants who completed the experiment and received a financial compensation, 37 participants signed up for the experiment on Prolific but did not start or complete it, and one participant was rejected as no data was available. The average time taken was 20 min. All data were collected on 6 June 2019.

### 5.1.2. Apparatus, stimuli and procedure

The trial procedure is depicted in figure 4. The apparatus and the online testing procedure was the same as in Experiment 1B. On each trial, participants saw four cards (arranged in a cross) in the centre of the screen (figure 4). On one-third of the trials, the vertical cards were purple and the two horizontal cards were grey. These were the non-gamble trials. On the other two-thirds of the trials, the horizontal cards were coloured (green, blue, orange or yellow) and the vertical cards were grey. These trials were the gamble trials on which participants could win or lose 5, 25 or 50 pence. The amounts (5, 25, 50 for gamble trials, and 0 for non-gamble trials) appeared in the middle of the screen, between the cards). The participants started (and ended) with an amount of £3. The rest of the procedure was the same as in Experiment 2. The experiment consisted of 288 trials which took approximately 20 min.

### 5.1.3. Analysis

As in the previous experiments, all data processing and analyses were completed with R [36]. The analyses were the same as in Experiment 2. Due to a failure in data online recording for one single trial in one participant, we excluded the trial in which the previous outcome was not known. Combined with the other exclusion criteria (see Experiment 1A), this resulted in a data exclusion of 2.9%.

**Table 4.** Inferential statistics Experiment 3. diff, difference; CI, confidence interval (95%); BF, Bayes Factor 10; $g_{av}$, Hedge's average $g$.

| | diff | lower CI | upper CI | d.f. | $t$ | $p$-value | BF | $g_{av}$ |
|---|---|---|---|---|---|---|---|---|
| **start RT** | | | | | | | | |
| non-gamble versus loss | 78.77 | 59.33 | 98.20 | 96 | 8.045 | <0.001 | $3.54 \times 10^9$ | 0.403 |
| non-gamble versus win | 34.82 | 16.42 | 53.22 | 96 | 3.756 | <0.001 | 68.38 | 0.172 |
| loss versus win | −43.95 | −59.28 | −28.62 | 96 | −5.691 | <0.001 | $9.18 \times 10^4$ | 0.213 |
| low versus medium | 7.18 | −7.02 | 21.34 | 96 | 1.004 | 0.318 | 0.18 | 0.036 |
| low versus high | −3.46 | −17.01 | 10.10 | 96 | −0.506 | 0.614 | 0.13 | 0.016 |
| medium versus high | −10.63 | −24.32 | 3.06 | 96 | −1.541 | 0.127 | 0.35 | 0.052 |
| **choice RT** | | | | | | | | |
| non-gamble versus loss | 23.83 | 11.73 | 35.93 | 96 | 3.910 | <0.001 | 112.71 | 0.109 |
| non-gamble versus win | −17.07 | −29.15 | −5.00 | 96 | −2.807 | 0.006 | 4.48 | 0.076 |
| loss versus win | −40.91 | −53.10 | −28.71 | 96 | −6.657 | <0.001 | $5.97 \times 10^6$ | 0.181 |

**Table 5.** Inferential statistics Experiment 3. diff, difference; CI, confidence interval (95%); BF, Bayes Factor 10; $g_{av}$, Hedge's average $g$.

| | diff | lower CI | upper CI | d.f. | $t$ | $p$-value | BF | $g_{av}$ |
|---|---|---|---|---|---|---|---|---|
| low win versus medium win | −10.96 | −30.05 | 8.12 | 96 | −1.140 | 0.257 | 0.21 | 0.05 |
| low win versus high win | −27.50 | −45.79 | −9.21 | 96 | −2.984 | 0.004 | 7.09 | 0.12 |
| medium win versus high win | −16.54 | −38.99 | 5.92 | 96 | −1.462 | 0.147 | 0.31 | 0.07 |
| low loss versus medium loss | 24.80 | 3.58 | 46.02 | 96 | 2.320 | 0.022 | 1.44 | 0.12 |
| low loss versus high loss | 20.48 | 1.78 | 39.19 | 96 | 2.174 | 0.032 | 1.06 | 0.09 |
| medium loss versus high loss | −4.32 | −24.31 | 15.67 | 96 | −0.429 | 0.669 | 0.12 | 0.02 |

## 5.2. Results and discussion

The start RT was influenced by the outcome of the previous trial: participants started the next trial faster after a loss ($M = 469$ ms; s.d. = 199 ms), than after a non-gamble trial ($M = 548$ ms; s.d. = 190 ms) or a win ($M = 513$ ms; s.d. = 213 ms). All differences were statistically significant (table 4, but see figure 5). This pattern is inconsistent with the orienting account as wins, losses and non-gambles occurred with the same frequency.

The 2 x 3 ANOVA with previous outcome and previous amount as independent variables revealed an interaction between previous outcome and previous amount, $F(2, 192) = 5.86$, $p = 0.003$. The main effect of amount was not significant, $F(2, 192) = 1.15$, $p = 0.315$. The main effect of previous outcome was significant, $F(1, 96) = 32.13$, $p < 0.001$.

To explore the interaction, we compared low, medium and high wins using *post hoc* $t$-tests (see table 5 for inferential statistics). After correction for multiple comparisons, we found a significant difference between trials following low wins ($M = 500$ ms; s.d. = 212 ms) and trials following high wins ($M = 527$ ms; s.d. = 233 ms). Trials following medium wins were numerically in between ($M = 511$ ms; s.d. = 214 ms), but did not differ significantly from the two other trial types. The difference between low and high wins is in line with the post-reinforcement pause (PRP) account. According to this account, latencies are prolonged after wins (the PRP), and the larger the win, the more pronounced the prolongation [44]. This is what we observed here as well.

We conducted the same *post hoc* $t$-tests with low, medium and high losses. After correction for multiple comparisons, we found no differences between trials following low losses ($M = 484$ ms; s.d. = 216 ms), medium losses ($M = 459$ ms; s.d. = 189 ms), or high losses ($M = 464$ ms; s.d. = 216 ms).

For the choice RT, we found that participants were faster after losses (M = 732 ms; s.d. = 217 ms) than after non-gamble trials (M = 756 ms; s.d. = 217 ms) or wins (M = 773 ms; s.d. = 232 ms). The difference between trials following non-gambles and wins was also significant.

In sum, we were able to replicate the findings of Verbruggen *et al.* [17] conceptually, finding faster initiation of the next trial after a loss, compared to a win or a non-gamble trial. The sequential effects appeared to be modulated by the previous amount though. Participants paused longer after a high win [33] compared to a medium win and a low win. Numerically, the post-loss speeding was largest in high and medium losses compared to low losses, but these differences were not significant after correcting for multiple comparisons.

# 6. General discussion

This study aimed to replicate and extend the findings of Verbruggen *et al.* [17]. We conducted four experiments using different tasks in which participants could win or lose points (that were converted into real money). We found that losses invigorated subsequent behaviour, which appears inconsistent with the post-error slowing effect [5,6] that is observed in many other tasks and that led researchers to conclude that suboptimal outcomes generally lead to response restraint and increased caution.

To investigate post-loss speeding, we first used the gambling task of the original study in the lab (Experiment 1A) and online (Experiment 1B). We were able to replicate the post-loss speeding effect in both experiments, finding that participants initiated the next trial faster after a loss than after a win or non-gamble trial. Additionally, we found that participants gambled more after a loss than after a win in Experiment 1B. This observation is in line with real-life observations showing that people continue gambling more after a loss than after a win [45]. Moreover, this observation is inconsistent with the idea that suboptimal outcomes lead to restraint or less risk-taking even in healthy participants [7].

In a second laboratory study, we used a variant of the 'doors task' [32], presenting three doors to the participants. On non-gamble trials (middle door coloured in green), participants simply had to press the space bar to continue the experiment (so they could not win or lose points). On gamble trials (indicated by the outer doors coloured in green), the participants had to guess under which door a reward was hidden (by pressing a left or right key). In this experiment, we did not observe a start RT difference between trials following a win or a loss (i.e. no post-loss speeding; note that post-loss speeding was observed in choice RTs). The start RT after gambles was shorter than start RT after non-gambles though. Therefore, we made some modifications to the task: we introduced a choice element on non-gambling trials, an irrelevant feature that might encourage participants to look for hidden rules, and we ran it online to increase the sample size. In this third experiment, we conceptually replicated the post-loss speeding effect.

In Experiment 3, we found numerically larger post-loss speeding with larger losses. Furthermore, we found that if the previous win was high, participants were even slower than after smaller wins. Moreover, the difference between high wins and high losses was larger than the difference between medium or low wins and losses. Therefore, we assume that both post-loss speeding and post-win slowing (i.e. post-reinforcement pause) contributed to the sequential effects. It seems that high amounts even enhance this post-reinforcement pause. Overall, the post-loss speeding and post-reinforcement pause findings seem to fit with Carver's [21] monitoring framework. This framework assumes a 'meta monitoring loop' in which the current state is compared with the goal state, and depending on how well the individual is doing in reaching this goal, behaviour is adjusted differently. For example, when the individual is currently doing well (e.g. winning a high amount of money on the current trial) in reaching the goal state (e.g. winning as much money is possible in the overall experiment), subsequent effort is reduced (e.g. pausing after a high win). However, if the individual is performing badly in reaching the goal state, subsequent effort is increased and behaviour is invigorated.

Combined, our findings appear inconsistent with standard cognitive control and orienting frameworks. Control (and related learning) accounts assume that suboptimal outcomes lead to more cautious behaviour [7,46]. Here, we observed speeding and increased risk taking. Importantly, the findings of Experiment 3 also allow us to rule out orienting accounts [10,11] as wins, losses and non-gambles occurred with equal probability. Instead, the main findings seem more in line with a 'frustrative non-reward' account. This account assumes that the omission of a reward in a usually rewarded situation becomes frustrating. This reward omission (as a negative event) leads to invigoration of behaviour [20] or as Frijda [24] puts it, 'impulsive action'.

Our findings are also in line with findings of post-error speeding in difficult tasks [47,48]. Williams *et al.* [48] found that participants tend to speed up when they realized they could not control the accuracy in this task. The authors assumed that participants become bored and less engaged with situations in which caution cannot lead to better performance. These findings are aligned with those of Dyson *et al.* [49] who found that participants only slowed down after a loss when participants could successfully apply a strategy to guide their choice. However, when no strategies could be applied, participants did not slow down but even sped up (for similar findings see [47]).

As a final observation, we found in all experiments that start RT was generally faster after gambles than non-gambles. Such general speeding might be caused by higher arousal in gamble trials compared to non-gamble trials. Consistent with this idea, previous studies found that arousal increased when there was uncertainty between a decision and a feedback [50,51]. Similarly, Robinson and colleagues [52] showed that this uncertainty-evoked arousal leads to invigoration of behaviour. Note that the arousal hypothesis might also account for another recent observation. Previous studies have shown that alcohol consumption can increase gambling behaviour [53–57]. Recently, Tobias-Webb *et al.* [58] showed that this influence can be mutual. More specific, they found that gambling could increase subsequent alcohol consumption. The outcome of the gambles did not appear to play a major role. Although speculative, the general speeding after gambles observed in the present study and the increase in alcohol consumption after gambling might be caused by similar (arousal-based) mechanisms.

# 7. Conclusion

In summary, we were able to replicate the findings by Verbruggen *et al.* [17], showing that in a gambling task, losses lead to more 'impulsive' behaviour on the subsequent trial compared to wins and non-gamble trials. Moreover, we were able to show that these findings cannot be explained by the orienting account and are not task-specific, but can be found more generally in win and loss situations.

Ethics. The study was approved by the local research ethics committee at the Faculty of Psychology and Educational Science of Ghent University. Informed consent was obtained.
Data accessibility. All raw data, processed data, experimental and analyses codes and materials can be found on https://osf.io/7xbth/ or with the doi: 10.17605/OSF.IO/7XBTH [37].
Authors' contributions. F.V. and C.E. designed the study. C.E., L.V. and Z.C. programmed the experiments. C.E., F.V. and Z.C. performed the analyses. C.E. and F.V. wrote the first draft and Z.C., L.V. and J.B. provided critical revisions on the manuscript. Note that Z.C. acted as 'co-pilot' for the whole project (see https://fredvbrug.github.io/openScience.html for more information).
Competing interests. The authors have no competing interests to declare.
Funding. This work was supported by an ERC Consolidator grant awarded to F.V. (European Union's Horizon 2020 research and innovation programme, grant agreement no. 769595).

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
