## [Reviewer comments · Royal Society Open Science]

Review History

RSOS-200090.R0 (Original submission)

Review form: Reviewer 1

Is the manuscript scientifically sound in its present form?

Yes

Are the interpretations and conclusions justified by the results?

Yes

Is the language acceptable?

Yes

Do you have any ethical concerns with this paper?

No

Have you any concerns about statistical analyses in this paper?

No

Recommendation?

Major revision is needed (please make suggestions in comments)

Comments to the Author(s)

In this submission Eben and colleagues attempt (successfully, in most cases) to replicate the post-loss speeding effect observed in risky choice by Verbruggen et al. (2017). The authors provide a more-or-less direct replication of the original study (Experiment 1) and additionally find evidence for post-loss speeding in a different task paradigm based on Hajcak's 'Doors' paradigm (Experiments 2 and 3). I assume that given the emphasis and scope of this journal, my review should not necessarily address the novelty of these results. Accordingly, while I found the experiments to be generally well designed and the analyses reasonable (thus making a potentially useful contribution to the literature on this topic) I have a number of comments that should help improve the presentation of this contribution and consequently its potential useful to future researchers.

- First, more conceptually, while this may not have been identified by the authors (or reviewers) as an issue in the original 2017 paper, why should we expect losses after a risky choice to relate at all to the notion of an "error" in cognitive control tasks? The authors interpret the originally observed post-loss speeding effects as evidence suggesting against the generality of post-error slowing. Is it a reasonable assumption that a decision-maker treats the negative outcome of a risky choice as an error? For example, imagine making a choice between \$1 with certainty and \$110 with a probability of 0.01 (and \$0 otherwise): while the expected values unequivocally the risky option in light of its low probability of winning, would a decision-maker treat this extremely likely \$0 outcome as an 'error'? In this framework, the correspondence between undesirable risky outcomes and errors in other tasks feels like a false equivalence.

- In my understanding the demographics of Prolific subjects are not well-characterized in the literature as its use is not (yet) as commonplace as say, Mechanical Turk or Crowdfunder. It would be useful to know where were these online subjects were located – and importantly can we assume they are they in English-speaking locales?

- It would be useful for the authors to report simple RTs by the *current* choice type (not just the previous choice type and outcome) for example in Table 1. Along these lines it would be helpful to know if the current choice type modulates the post-lost speeding effect – i.e. is this 'start' or 'choice' effect larger on current gamble versus sure-thing trials?

- The lead-in to Experiment 2 asserts that in gambling trials, winning is less likely than losing. To me this is not evident as there were a range of outcome probabilities examined (some of which are greater than 0.5). Am I missing something here?

- The lead-in to Experiment 3 (page 18), at the end of the Experiment 2 Results, simply states "we made some modifications to the procedure." It would be tremendously useful here to provide some logic here guiding readers intuitions about changes to the Doors procedure which might permit observation of the post-loss speeding effect.

- What evidence is there (in terms of a power analysis or some other inferential technique) suggesting that Experiment 2 was not sufficiently powered to detect an effect?

- Why is the first-half / second-half analysis of Experiment 2 first reported in Experiment 3, and without any inferential statistics?

- I found that the clarify and grammatical correctness of the writing dropped precipitously around Experiment 3 (and thereafter). I would suggest some more careful proofreading/re-writing of the revision.

- Figure 5 would benefit from labeling the experiments within each panel.

Review form: Reviewer 2 (Haiyan Wu)

Is the manuscript scientifically sound in its present form?

Yes

Are the interpretations and conclusions justified by the results?

Yes

Is the language acceptable?

Yes

Do you have any ethical concerns with this paper?

No

Have you any concerns about statistical analyses in this paper?

Yes

Recommendation?

Major revision is needed (please make suggestions in comments)

Comments to the Author(s)

This manuscript provided a replication of the previous study, based on 4 experiments from both lab and online studies, to replicate the effect of faster initiation of the next trial after a loss, compared to a win or a non-gamble trial.

The authors provided open data and code of the study which is appreciated as a practice of open science. Various data from different tasks have been collected to test the hypothesis. I respect the authors' efforts; however, some points of concern are listed below.

1) As for the online study, the amount of data loss or uncompleted data is very high. It should be noted that the task engagement problem as it looks like a random choice. Moreover, the gender ratio of experiment 2 should also be tested;

2) I doubt whether other potential factors that may mix with the effect, e.g., the probability of gambling post-loss choice. Can the author clearly address the hypothesis or account they support?

Review form: Reviewer 3 (Luke Clark)

Is the manuscript scientifically sound in its present form?

Yes

Are the interpretations and conclusions justified by the results?

Yes

Is the language acceptable?

Yes

Do you have any ethical concerns with this paper?

No

Have you any concerns about statistical analyses in this paper?

No

Recommendation?

Accept with minor revision (please list in comments)

Comments to the Author(s)

This is a well-written and thorough report of 4 experiments (1a and 1b, 2 and 3) looking at a psychological effect of speeding following losing outcomes. The findings are interesting in complementing the well-established effect of the 'post reinforcement pause' (slower responses following wins) from learning theory, and more importantly, the authors construct an interesting theoretical framework for studying their effect in relation to well-known 'post error slowing' phenomenon in the cognitive control literature. Their findings indicate that post error slowing cannot be a general phenomenon.

I have few comments on this manuscript, I commend the authors for a very tidy job. As a general comment, the authors report the Start RT and the Choice RT in each experiment, and their effects pertain primarily to the Start RT. It makes sense to me that frustrative mechanisms would be particularly powerful on StartRT, but in my view (which may be limited to certain tasks), conventional psychological experiments often proceed automatically from one outcome to the next trial, ie. without requiring a formal 'start' response, and would thus overlook StartRT. How does this shape their theoretical interpretation regarding post-error slowing? Are there specific past experiments of post-error slowing that find the effect when distinguishing StartRT and ChoiceRT? Or is their main conclusion more methodological, that more experiments should insert a StartRT?

As a follow-on point, do the authors find it surprising that they can discern any sequential effects at ChoiceRT when variable delays have already been imposed via the StartRT?

Minor comments

Pg 3, line 55 'orient attention away' – away from what?

Pg 9, line 47: the authors specify the exclusion threshold per participant, but do not say for expt 1a how many participants were excluded. (If none, please specify)

In expts 1a and 1b, some readers may expect an 'omnibus' statistical test with a repeated-measures ANOVA, before the series of t tests. In the absence of a preregistration plan, it may be helpful to quickly explain why the ANOVA was not considered useful. (Especially as expts 2 and 3 do report the ANOVA to test the modulation by magnitude)

Ref to Saunders & Inzlicht (2015) is missing from bibliography.

Decision letter (RSOS-200090.R0)

10-Mar-2020

Dear Miss Eben

On behalf of the Editors, I am pleased to inform you that your Manuscript RSOS-200090 entitled "A direct and conceptual replication of post-loss speeding when gambling" has been accepted for publication in Royal Society Open Science subject to minor revision in accordance with the referee suggestions. Please find the referees' comments at the end of this email.

The reviewers and handling editors have recommended publication, but also suggest some minor revisions to your manuscript. Therefore, I invite you to respond to the comments and revise your manuscript.

- Ethics statement

- Data accessibility

If you wish to submit your supporting data or code to Dryad (<http://datadryad.org/>), or modify your current submission to dryad, please use the following link:
<http://datadryad.org/submit?journalID=RSOS&manu=RSOS-200090>

- Competing interests

- Authors' contributions

- Acknowledgements

- Funding statement

Because the schedule for publication is very tight, it is a condition of publication that you submit the revised version of your manuscript before 19-Mar-2020. Please note that the revision deadline will expire at 00.00am on this date. If you do not think you will be able to meet this date please let me know immediately.

To revise your manuscript, log into <https://mc.manuscriptcentral.com/rsos> and enter your

Author Centre, where you will find your manuscript title listed under "Manuscripts with Decisions". Under "Actions," click on "Create a Revision." You will be unable to make your revisions on the originally submitted version of the manuscript. Instead, revise your manuscript and upload a new version through your Author Centre.

If your manuscript is newly submitted and subsequently accepted for publication, you will be asked to pay the article processing charge, unless you request a waiver and this is approved by Royal Society Publishing. You can find out more about the charges at <https://royalsocietypublishing.org/rsos/charges>. Should you have any queries, please contact opscience@royalsociety.org.

on behalf of Dr Anastasia Christakou (Associate Editor) and Essi Viding (Subject Editor)
 openscience@royalsociety.org

Associate Editor Comments to Author (Dr Anastasia Christakou):

Thank you for the opportunity to read this manuscript. You will see that reviews are on the whole constructive and receptive of the manuscript aims.

In preparing your response, I would like to highlight in particular the conceptual question raised by one of the reviewers about the putative relationship between processing losses following a risky choice and the notion of error processing. Dealing with this inherent assumption is important for theoretical clarity.

I will also highlight the question, raised by another reviewer, about clarifying the relationship of the sequential effects reported at ChoiceRT with the delay variability imposed by the StartRT. I look forward to your response to the reviews, which I am sure will be as thorough and well presented as the manuscript itself.

Reviewer comments to Author:

Reviewer: 1

Comments to the Author(s)

In this submission Eben and colleagues attempt (successfully, in most cases) to replicate the post-loss speeding effect observed in risky choice by Verbruggen et al. (2017). The authors provide a more-or-less direct replication of the original study (Experiment 1) and additionally find evidence for post-loss speeding in a different task paradigm based on Hajcak's 'Doors' paradigm (Experiments 2 and 3). I assume that given the emphasis and scope of this journal, my review should not necessarily address the novelty of these results. Accordingly, while I found the experiments to be generally well designed and the analyses reasonable (thus making a potentially useful contribution to the literature on this topic) I have a number of comments that should help improve the presentation of this contribution and consequently its potential usefulness to future researchers.

- First, more conceptually, while this may not have been identified by the authors (or reviewers) as an issue in the original 2017 paper, why should we expect losses after a risky choice to relate at all to the notion of an "error" in cognitive control tasks? The authors interpret the originally observed post-loss speeding effects as evidence suggesting against the generality of post-error slowing. Is it a reasonable assumption that a decision-maker treats the negative outcome of a risky choice as an error? For example, imagine making a choice between \$1 with certainty and \$110 with a probability of 0.01 (and \$0 otherwise): while the expected values unequivocally favor the risky option in light of its low probability of winning, would a decision-maker treat this extremely likely \$0 outcome as an 'error'? In this framework, the correspondence between undesirable risky outcomes and errors in other tasks feels like a false equivalence.

- In my understanding the demographics of Prolific subjects are not well-characterized in the literature as its use is not (yet) as commonplace as say, Mechanical Turk or Crowdflower. It

would be useful to know where were these online subjects were located – and importantly can we assume they are they in English-speaking locales?

- It would be useful for the authors to report simple RTs by the *current* choice type (not just the previous choice type and outcome) for example in Table 1. Along these lines it would be helpful to know if the current choice type modulates the post-lost speeding effect – i.e. is this ‘start’ or ‘choice’ effect larger on current gamble versus sure-thing trials?

- The lead-in to Experiment 2 asserts that in gambling trials, winning is less likely than losing. To me this is not evident as there were a range of outcome probabilities examined (some of which are greater than 0.5). Am I missing something here?

- The lead-in to Experiment 3 (page 18), at the end of the Experiment 2 Results, simply states “we made some modifications to the procedure.” It would be tremendously useful here to provide some logic here guiding readers intuitions about changes to the Doors procedure which might permit observation of the post-loss speeding effect.

- What evidence is there (in terms of a power analysis or some other inferential technique) suggesting that Experiment 2 was not sufficiently powered to detect an effect?

- Why is the first-half / second-half analysis of Experiment 2 first reported in Experiment 3, and without any inferential statistics?

- I found that the clarify and grammatical correctness of the writing dropped precipitously around Experiment 3 (and thereafter). I would suggest some more careful proofreading/re-writing of the revision.

- Figure 5 would benefit from labeling the experiments within each panel.

Reviewer: 2

Comments to the Author(s)

This manuscript provided a replication of the previous study, based on 4 experiments from both lab and online studies, to replicate the effect of faster initiation of the next trial after a loss, compared to a win or a non-gamble trial.

The authors provided open data and code of the study which is appreciated as a practice of open science. Various data from different tasks have been collected to test the hypothesis. I respect the authors’ efforts; however, some points of concern are listed below.

1) As for the online study, the amount of data loss or uncompleted data is very high. It should be noted that the task engagement problem as it looks like a random choice. Moreover, the gender ratio of experiment 2 should also be tested;

2) I doubt whether other potential factors that may mix with the effect, e.g., the probability of gambling post-loss choice. Can the author clearly address the hypothesis or account they support?

Reviewer: 3

Comments to the Author(s)

This is a well-written and thorough report of 4 experiments (1a and 1b, 2 and 3) looking at a psychological effect of speeding following losing outcomes. The findings are interesting in complementing the well-established effect of the ‘post reinforcement pause’ (slower responses following wins) from learning theory, and more importantly, the authors construct an interesting theoretical framework for studying their effect in relation to well-known ‘post error slowing’

phenomenon in the cognitive control literature. Their findings indicate that post error slowing cannot be a general phenomenon.

I have few comments on this manuscript, I commend the authors for a very tidy job. As a general comment, the authors report the Start RT and the Choice RT in each experiment, and their effects pertain primarily to the Start RT. It makes sense to me that frustrative mechanisms would be particularly powerful on StartRT, but in my view (which may be limited to certain tasks), conventional psychological experiments often proceed automatically from one outcome to the next trial, ie. without requiring a formal 'start' response, and would thus overlook StartRT. How does this shape their theoretical interpretation regarding post-error slowing? Are there specific past experiments of post-error slowing that find the effect when distinguishing StartRT and ChoiceRT? Or is their main conclusion more methodological, that more experiments should insert a StartRT?

As a follow-on point, do the authors find it surprising that they can discern any sequential effects at ChoiceRT when variable delays have already been imposed via the StartRT?

Minor comments

Pg 3, line 55 'orient attention away' – away from what?

Pg 9, line 47: the authors specify the exclusion threshold per participant, but do not say for expt 1a how many participants were excluded. (If none, please specify)

In expts 1a and 1b, some readers may expect an 'omnibus' statistical test with a repeated-measures ANOVA, before the series of t tests. In the absence of a preregistration plan, it may be helpful to quickly explain why the ANOVA was not considered useful. (Especially as expts 2 and 3 do report the ANOVA to test the modulation by magnitude)

Ref to Saunders & Inzlicht (2015) is missing from bibliography.

Author's Response to Decision Letter for (RSOS-200090.R0)

See Appendix A.

RSOS-200090.R1 (Revision)

Review form: Reviewer 1

Is the manuscript scientifically sound in its present form?

Yes

Are the interpretations and conclusions justified by the results?

Yes

Is the language acceptable?

Yes

Do you have any ethical concerns with this paper?

No

Have you any concerns about statistical analyses in this paper?

No

Recommendation?

Accept with minor revision (please list in comments)

Comments to the Author(s)

The authors have addressed many of my concerns but the following issues still remain:

- I appreciate the author's inclusion of rationale for analyzing errors and losses similarly (p 4). But can the authors do this without relying so heavily upon neuroscientific reverse inference (e.g. equating the two processes based on similar patterns of brain activation or fEMG responses)?
- Demographic information, to the extent you have it, for your online studies, should be reported in the manuscript because it is important for future researchers who may use this paradigm.

Decision letter (RSOS-200090.R1)

Dear Miss Eben:

On behalf of the Editors, I am pleased to inform you that your Manuscript RSOS-200090.R1 entitled "A direct and conceptual replication of post-loss speeding when gambling" has been accepted for publication in Royal Society Open Science subject to minor revision in accordance with the referee suggestions. Please find the referees' comments at the end of this email.

The reviewers and Subject Editor have recommended publication, but also suggest some minor revisions to your manuscript. Therefore, I invite you to respond to the comments and revise your manuscript.

- Ethics statement

- Data accessibility

<http://datadryad.org/submit?journalID=RSOS&manu=RSOS-200090.R1>

- Competing interests

- Authors' contributions

- Acknowledgements

- Funding statement

Because the schedule for publication is very tight, it is a condition of publication that you submit the revised version of your manuscript before 01-May-2020. Please note that the revision deadline will expire at 00.00am on this date. If you do not think you will be able to meet this date please let me know immediately.

- 1) A text file of the manuscript (tex, txt, rtf, docx or doc), references, tables (including captions) and figure captions. Do not upload a PDF as your "Main Document".
- 2) A separate electronic file of each figure (EPS or print-quality PDF preferred (either format should be produced directly from original creation package), or original software format)

- 3) Included a 100 word media summary of your paper when requested at submission. Please ensure you have entered correct contact details (email, institution and telephone) in your user account
- 4) Included the raw data to support the claims made in your paper. You can either include your data as electronic supplementary material or upload to a repository and include the relevant doi within your manuscript
- 5) All supplementary materials accompanying an accepted article will be treated as in their final form. Note that the Royal Society will neither edit nor typeset supplementary material and it will be hosted as provided. Please ensure that the supplementary material includes the paper details where possible (authors, article title, journal name).

on behalf of Dr Anastasia Christakou (Associate Editor)
openscience@royalsociety.org

Associate Editor Comments to Author (Dr Anastasia Christakou):

Thank you for addressing the reviewers' comments so thoroughly. You will see that the critique remains regarding your explanation for treating errors and losses as equivalent events. You will have seen from the initial review cycle that I share the Reviewer's concern, so I would emphasise their suggestion to moderate the importance of relying on reverse inference (equivalent neural signatures do not necessarily equate equivalent psychological processes). This moderation may look like a theoretical expansion on the putative equivalence between losses and errors (this seems unnecessary to me), or simply a statement of caution in treating them as equivalent based on indirect physiological evidence alone. I also agree with the reviewer that any sample demographics, if acquired, should be explicitly reported at least in aggregate form. I look forward to receiving the final version of your manuscript.

Reviewer comments to Author:
 Reviewer: 1

Comments to the Author(s)
 The authors have addressed many of my concerns but the following issues still remain:

- I appreciate the author's inclusion of rationale for analyzing errors and losses similarly (p 4). But can the authors do this without relying so heavily upon neuroscientific reverse inference (e.g. equating the two processes based on similar patterns of brain activation or fEMG responses)?

- Demographic information, to the extent you have it, for your online studies, should be reported in the manuscript because it is important for future researchers who may use this paradigm.

Author's Response to Decision Letter for (RSOS-200090.R1)

See Appendix B.

Decision letter (RSOS-200090.R2)

Dear Miss Eben,

It is a pleasure to accept your manuscript entitled "A direct and conceptual replication of post-loss speeding when gambling" in its current form for publication in Royal Society Open Science.

on behalf of Dr Anastasia Christakou (Associate Editor)
openscience@royalsociety.org

Associate Editor Comments to Author (Dr Anastasia Christakou):

Many thanks for your considerate and thorough response to the review process, and congratulations on the paper.

Appendix A

Dear Dr. Christakou,

This letter accompanies the revision of our manuscript entitled “A direct and conceptual replication of post-loss speeding when gambling”. We are grateful for the comments by you and the reviewers, and changed our manuscript accordingly.

A complete and detailed response to the comments of the reviewers can be found below. In your letter, you highlighted two comments, namely (1) ‘the putative relationship between processing losses following a risky choice and the notion of error processing’, and (2) ‘the relationship of the sequential effects reported at the Choice RT with the delay variability imposed by the Start RT’.

With regards to your first comment, the existing literature shows similar neural and facial EMG activation patterns in response to errors and losses. Based on these findings, it has been argued that people respond to various types of sub-optimal outcomes in a similar fashion. We added these observations to our introduction on p. 4 in order to clarify our theoretical premise.

With regards to the second comment, we want to emphasise that the start RT is our main dependent variable as the choice RT (at least in Experiment 1a and 1b) is always ‘contaminated’ by the choice. More specifically, our previous findings indicate that for most participants, choice RT is generally longer for gambles than for non-gambles. As outcome of the previous trial also influences participants’ tendency to gamble on the current trial (i.e. they tend to gamble less after wins and more after losses), this complicates the interpretation of choice RT (as different effects might interact or even counteract each other). Here we are mainly interested in action vigour directly following a sub-optimal (potentially frustrative) event; therefore, the start RT is our main dependent variable of interest because it is not influenced by choice. We do report results on choice RT for completeness though.

Related to this, the reviewer was wondering if we were surprised that choice RT was influenced, even though variable delays had ‘already been imposed via the StartRT’. We were not surprised by this as we know from previous work that sequential effects can influence multiple trials/actions. Furthermore, our previous work has shown that start RT in the gambling task was influenced by the outcome of the previous gamble even when an extra task (with two extra responses) was inserted between the two gambling games.

A detailed point-to-point response to the other questions is provided below. Where appropriate a reference to the respective changes in the manuscript is given.

Yours sincerely,
Charlotte Eben
Zhang Chen
Luc Vermeulen
Joël Billieux
Frederick Verbruggen

Reviewer 1:

First, more conceptually, while this may not have been identified by the authors (or reviewers) as an issue in the original 2017 paper, why should we expect losses after a risky choice to relate at all to the notion of an “error” in cognitive control tasks? The authors interpret the originally observed post-loss speeding effects as evidence suggesting against the generality of post-error slowing. Is it a reasonable assumption that a decision-maker treats the negative outcome of a risky choice as an error? For example, imagine making a choice between \$1 with certainty and \$110 with a probability of 0.01 (and \$0 otherwise): while the expected values unequivocally the risky option in light of its low probability of winning, would a decision-maker treat this extremely likely \$0 outcome as an ‘error’? In this framework, the correspondence between undesirable risky outcomes and errors in other tasks feels like a false equivalence.

Reply: Indeed, at first sight errors and losses seem to be different. However, previous work suggests that similar neural networks (typically associated with performance monitoring) are activated by errors in choice tasks and losses in a gambling tasks (Nieuwenhuis, S., Yeung, N., Holroyd, C. B., Schurger, A., & Cohen, J. D. (2004). Sensitivity of electrophysiological activity from medial frontal cortex to utilitarian and performance feedback. *Cerebral Cortex*, 14(7), 741-747). Studies measuring facial EMG activity (often used as a marker of emotional responses) after errors and losses also indicate overlap (e.g. Lindström, B. R., Mattsson-Märn, I. B., Golkar, A., & Olsson, A. (2013). In your face: Risk of punishment enhances cognitive control and error-related activity in the corrugator supercillii muscle. *PLOS one*, 8(6); Elkins-Brown, N., Saunders, B., & Inzlicht, M. (2016). Error-related electromyographic activity over the corrugator supercillii is associated with neural performance monitoring. *Psychophysiology*, 53(2), 159-170.). Therefore, consistent with previous proposals, our premise is that losses and errors (which are both sub-optimal or undesirable outcomes) are processed in a similar way. We added this explanation and the reference on p.4 in the introduction.

In my understanding the demographics of Prolific subjects are not well-characterized in the literature as its use is not (yet) as commonplace as say, Mechanical Turk or Crowdfunder. It would be useful to know where were these online subjects were located—and importantly can we assume they are they in English-speaking locales?

Reply: In theory, participants from all over the world can sign up for our online studies via Prolific. However, as our instructions were in English (in all experiments, including the lab-based studies), we added knowledge of English as a criterion for Experiment 1B. In this study, 38% of participants were native English, 16% Polish, 7% Portuguese, 6% Italian, 5% German, 4% Estonian, 4% Greek, 4% Spanish, 2% Dutch, 2%, Russian, 1% Czech, 1% French, 1% Indonesian, 1% Japanese, 1% Serbian, 1% Slovenian, 1% Swedish and 1% Tagalog-Filipino. For the second online experiment, we decided to only test a native English speaking sample.

*It would be useful for the authors to report simple RTs by the *current* choice type (not just the previous choice type and outcome) for example in Table 1. Along these lines it would be helpful to know if the current choice type modulates the post-lost speeding effect—i.e. is this ‘start’ or ‘choice’ effect larger on current gamble versus sure-thing trials?*

Reply: The current choice cannot have any influence on the start RT as the participants do not know the choice options yet. With regards to the choice RT, it is indeed influenced by the choice itself. Verbruggen et al. 2017 already showed that (on average) choice RT was longer when participants selected the gamble vs. the non-gamble. As you can see in the figure below, we were able to replicate this pattern. However, we were mainly interested in sequential effects in this study. For many participants, trial numbers were too low to (properly) examine how current choice and outcome of the previous trial influence choice RT. But to give a general impression of what the pattern might look like, below we have plotted the group averages for this interaction as well.

Experiment 1A:

Experiment 1B:

The lead-in to Experiment 2 asserts that in gambling trials, winning is less likely than losing. To me this is not evident as there were a range of outcome probabilities examined (some of which are greater than 0.5). Am I missing something here?

Reply: It is true that this only holds when participants gamble on most of the

trials (regardless of the probability of winning). In that case we have one out of four options with a probability of winning larger than .50, one option with a probability of .50 and two options with a probability of winning lower than .50 (resulting in an overall probability of winning of 43.75%). Note that the actual probability to win was 46% across both experiments. We specified that in the manuscript on p. 15.

The lead-in to Experiment 3 (page 18), at the end of the Experiment 2 Results, simply states “we made some modifications to the procedure.” It would be tremendously useful here to provide some logic here guiding readers intuitions about changes to the Doors procedure which might permit observation of the post-loss speeding effect.

Reply: In order to not repeat ourselves, we decided to provide the logic and modifications in the introduction of Experiment 3 which can be found on p. 19. We added that the modifications are discussed in the next section.

What evidence is there (in terms of a power analysis or some other inferential technique) suggesting that Experiment 2 was not sufficiently powered to detect an effect?

Reply: The effect size for the contrast between wins and losses was very small ($g_{av} = .082$) in Experiment 2. Note that the effect size of this post-loss speeding effect in Experiment 3 was low as well ($g_{av} = .213$), but here, our sample size was much larger). Please note, that we deleted the paragraph about the power issue in the manuscript as we decided it does not add much.

Why is the first-half / second-half analysis of Experiment 2 first reported in Experiment 3, and without any inferential statistics?

Reply: This analysis was not planned a priori. We conducted this analysis only to test the hypothesis that participants might have given up after realising that they did not have any influence. However there are not enough trials per condition to allow any (proper) inferential statistics.

I found that the clarify and grammatical correctness of the writing dropped precipitously around Experiment 3 (and thereafter). I would suggest some more careful proofreading/re-writing of the revision.

Reply: We corrected that.

Figure 5 would benefit from labeling the experiments within each panel.

Reply: We corrected that.

Reviewer 2:

As for the online study, the amount of data loss or uncompleted data is very high. It should be noted that the task engagement problem as it looks like a random choice. Moreover, the gender ratio of experiment 2 should also be tested

Reply: Please note that the final number included all participants who initially signed up for the experiment. It seems a common practice on Prolific to sign up for studies as fast as possible (especially when the studies give more than a couple of pence for a couple of minutes) to save a spot and then figuring out if they actually want to participate (this is what we experienced so far, also through messages from the participants directly). Thus, some participants never actually started the experiment. We corrected that in the sections ‘participants’ in Experiment 1B and Experiment 3 on the pages 12 and 20.

With regard to the gender ratio: Yes, there were more females than males, but this is common for most lab-based studies that use psychology students. Note that the ratio was more balanced in the online studies.

I doubt whether other potential factors that may mix with the effect, e.g., the probability of gambling post-loss choice. Can the author clearly address the hypothesis or account they support?

Reply: Indeed, the choice RT is contaminated by the choice (probability of gambling) itself, so we used the start RT (for further discussion, see our response to similar questions by Reviewers 1 and 3).

Reviewer 3:

As a general comment, the authors report the Start RT and the Choice RT in each experiment, and their effects pertain primarily to the Start RT. It makes sense to me that frustrative mechanisms would be particularly powerful on StartRT, but in my view (which may be limited to certain tasks), conventional psychological experiments often proceed automatically from one outcome to the next trial, ie. without requiring a formal 'start' response, and would thus overlook StartRT. How does this shape their theoretical interpretation regarding post-error slowing? Are there specific past experiments of post-error slowing that find the effect when distinguishing StartRT and ChoiceRT? Or is their main conclusion more methodological, that more experiments should insert a StartRT?

Reply: To our knowledge, no studies have been published yet on sequential effects of errors on start RT in self-paced choice tasks. However, Verbruggen and colleagues (2017) found shorter start RT after an error in a decision making task that was intermixed with the gambling task. Furthermore, some unpublished work from our lab indicates that start RT effects can be observed in non-gambling contexts as well (at least under some circumstances). Whether researchers should make tasks self-paced and measure start RT depends on the research question. For example, if early appraisal or orienting effects (e.g. Wessel, 2018) are not of interest, it might be more useful to have fixed intertrial intervals that are sufficiently long.

As a follow-on point, do the authors find it surprising that they can discern any sequential effects at ChoiceRT when variable delays have already been imposed via the StartRT?

Reply: We were not surprised that choice RT was influenced, even though variable delays had 'already been imposed via the StartRT'. After all, we know from previous work that sequential effects can influence multiple trials/actions. Furthermore, our previous work has shown that start RT in the gambling task was influenced by the outcome of the previous gamble even when an extra task (with two extra responses) was inserted between the two gambling games.

Minor comments Pg 3, line 55 'orient attention away' – away from what?

Reply: Away from the task; we now specify that on p. 3.

Pg 9, line 47: the authors specify the exclusion threshold per participant, but do not say for expt 1a how many participants were excluded. (If none, please specify)

Reply: The number of excluded participants can be found in the section 'participants' for each section.

In expts 1a and 1b, some readers may expect an 'omnibus' statistical test with a repeated-measures ANOVA, before the series of t tests. In the absence of a

preregistration plan, it may be helpful to quickly explain why the ANOVA was not considered useful. (Especially as expts 2 and 3 do report the ANOVA to test the modulation by magnitude)

Reply: As we were trying to replicate the previous results, and given our main research question, we were mostly interested in the differences of losses and wins with the non-gambling baseline (i.e. the omnibus ANOVA cannot inform us whether losses, wins, or both are influencing action). Therefore, we directly conducted the (planned) pairwise comparisons.

Ref to Saunders & Inzlicht (2015) is missing from bibliography.

Reply: Thanks for pointing out this omission, we corrected that.

Appendix B

Dear Dr. Christakou,

This letter accompanies the revision of our manuscript entitled “A direct and conceptual replication of post-loss speeding when gambling”. Thank you for the comments by you and the reviewer, we changed our manuscript accordingly.

We fully agree that one should be careful when interpreting neural and psychophysiological data. However, we merely wanted to point out that (based on such data) it has been argued in the past that different types of suboptimal outcomes (such as errors and losses, but also various types of conflicts) are treated in a similar way. To address the concern, in the present version of the manuscript, we have added a statement of caution and refer to the problem of reverse inference on p. 2 in the last paragraph: “Thus, based on such findings, it has been argued that losses and errors (which are both sub-optimal or undesirable outcomes) are processed in a similar way (e.g. Nieuwenhuis, 2004). However, one should be aware of potential reverse inference problems (see e.g. Poldrack, 2006). Furthermore, despite this (purported) overlap, speeding (instead of slowing) has been observed after suboptimal outcomes (losses) in gambling tasks. For example, Verbruggen and colleagues (2017) used a simple gambling task in which participants choose between a non-gamble option with a guaranteed (low) amount of points and a gambling option with a higher amount of points but a lower probability of winning.”.

Moreover, we added the demographic information we have about the first language (Experiment 1B; retrieved from the data Prolific provides) nationality of the participants in our online experiments.

For Experiment 1B, the following information is provided on p.7: “In theory, participants from all over the world can sign up for our online studies via Prolific. However, as our instructions were in English (in all experiments, including the lab-based studies), we added knowledge of English as a criterion for Experiment 1B. For the first language in this study Prolific provided us with the following data: 38% of participants were native English, 16% Polish, 7% Portuguese, 6% Italian, 5% German, 4% Estonian, 4% Greek, 4% Spanish, 2% Dutch, 2%, Russian, 1% Czech, 1% French, 1% Indonesian, 1% Japanese, 1% Serbian, 1% Slovenian, 1% Swedish and 1% Tagalog-Filipino speaking. For the nationality in this study, 22% were English (UK) participants, 17 % Polish, 8 % Portuguese, 6% Italian, 6 % Australian, 4 % Greek, 4 % Estonian, 3 % German, 3 % Mexican, 2 % Austrian, 2 % Canadian, 2 % Latvian, 2 % Finnish, 2 % American, 2 % New Zealand and 1 % each Belgian, Czech, Dutch, French, Indonesian, Iranian, Israeli, Japanese, Luxembourgish, Philippines, Serbian, Slovenian, Spanish, Swedish and Vietnamese. ”

For Experiment 3, the following information is provided on p.12: “In the experiment we decided to only test a native English speaking sample. In this sample were 63 % English (UK) participants, 16 % American, 10 % Australian, 4 % Canadian, 3 % New Zealand, and 1 % each Iraqi, Irish, Philippines and

Spanish.”

We thank you for considering our manuscript for publication and look forward to hearing from you in due course.

Yours sincerely,
Charlotte Eben
Zhang Chen
Luc Vermeyleylen
Joël Billieux
Frederick Verbruggen